# Key Aroma Compounds of Dark Chocolates Differing in Organoleptic Properties: A GC-O Comparative Study

**DOI:** 10.3390/molecules25081809

**Published:** 2020-04-15

**Authors:** Zoé Deuscher, Karine Gourrat, Marie Repoux, Renaud Boulanger, Hélène Labouré, Jean-Luc Le Quéré

**Affiliations:** 1Centre des Sciences du Goût et de l’Alimentation (CSGA), AgroSup Dijon, CNRS, INRAE, Université de Bourgogne Franche-Comté, F-21000 Dijon, France; 2CIRAD, UMR Qualisud, F-34398 Montpellier, France; 3ChemoSens Platform, CSGA, F-21000 Dijon, France; 4Valrhona, 14 av. du Président Roosevelet, F-26602 Tain l’Hermitage, France; 5Qualisud, Univ Montpellier, CIRAD, Montpellier SupAgro, Univ d’Avignon, Univ de La Réunion, F-34000 Montpellier, France

**Keywords:** dark chocolate, key odorant, key aroma, gas chromatography-olfactometry (GC-O), comparative detection frequency analysis (cDFA), nasal impact frequency (NIF), correspondence analysis (CA), hierarchical cluster analysis (HCA), heatmap

## Abstract

Dark chocolate samples were previously classified into four sensory categories. The classification was modelled based on volatile compounds analyzed by direct introduction mass spectrometry of the chocolates’ headspace. The purpose of the study was to identify the most discriminant odor-active compounds that should characterize the four sensory categories. To address the problem, a gas chromatography-olfactometry (GC-O) study was conducted by 12 assessors using a comparative detection frequency analysis (cDFA) approach on 12 exemplary samples. A nasal impact frequency (NIF) difference threshold combined with a statistical approach (Khi² test on k proportions) revealed 38 discriminative key odorants able to differentiate the samples and to characterize the sensory categories. A heatmap emphasized the 19 most discriminant key odorants, among which heterocyclic molecules (furanones, pyranones, lactones, one pyrrole, and one pyrazine) played a prominent role with secondary alcohols, acids, and esters. The initial sensory classes were retrieved using the discriminant key volatiles in a correspondence analysis (CA) and a hierarchical cluster analysis (HCA). Among the 38 discriminant key odorants, although previously identified in cocoa products, 21 were formally described for the first time as key aroma compounds of dark chocolate. Moreover, 13 key odorants were described for the first time in a cocoa product.

## 1. Introduction

Dark chocolate may contain 35%, and up to 85%–99% for high cocoa content samples, of the ingredients originating from cocoa (cocoa solids and cocoa butter). The appreciation of dark chocolate is mainly related to its sensory properties, which are greatly influenced by the cocoa beans’ aroma and by the complex manufacturing process [1] that gives rise to the final chocolate product. The volatile composition of cocoa beans and of the resulting dark chocolate has been the subject of many gas chromatography-mass spectrometry (GC-MS) studies, with the aim of characterizing the (i) chocolate quality attributes, (ii) variety and origin of cocoa beans, and (iii) the process, including the fermentation and drying of cocoa beans, roasting, and conching.

Thus, quality attributes of dark chocolate produced from Vietnamese cocoa were recently investigated [2], as well as the influence of the cocoa variety on the fermentation step studied [3,4,5], influence of the cocoa origin on the cocoa flavor examined [6,7,8,9], and the link between the origin and process searched for [10]. The process, starting with fermentation [11,12,13,14,15,16] and followed by drying [17] and roasting [18,19,20,21,22,23] steps, has been the subject of many studies. Specialized reviews and treatises gathered such knowledge [1,24,25,26].

Among these investigations, studies aiming to identify the only aroma-active compounds in dark chocolate (the dark chocolate key aroma compounds) are scarce. Nevertheless, some important gas chromatography-olfactometry (GC-O) studies, completed by GC-MS, allowed identification of the major dark chocolate key aroma compounds. Thus, the influence of conching was examined, with the aim of identifying the key odorant compounds in dark chocolate [27]. Aroma-active compounds in dark and milk chocolate in relation to sensory perception were investigated [28], and recently, key aroma compounds in two commercial dark chocolates with high cocoa contents were characterized [29]. As already cited, the major results are reported in reviews on cocoa and cocoa products [24,25,26] as well as in a specialized treatise dedicated to chocolate [1].

Direct injection mass spectrometry (DIMS) methods, such as proton transfer reaction mass spectrometry (PTR-MS), have been used in some studies conducted on dark chocolates. Their objective was generally to classify the chocolates according to the variety and/or the origin of the transformed cocoa beans by measuring and comparing their relative volatile organic compounds (VOCs) patterns [30,31]. Investigations into the relationships between VOC patterns obtained using a DIMS method and the organoleptic properties of foodstuffs obtained as sensory profiles measured by a panel are rather scarce [32]. Dark chocolates of diverse cocoa origins and cultivars, but manufactured using the same standard process at the pilot level at an industrial plant, could be classified into four sensory categories [32]. This classification was based on a quantitative descriptive analysis (QDA) protocol conducted by an internal expert panel: The panelists rated 36 flavor attributes, among which 33 were aroma descriptors. The classification into the four sensory categories is, in this case, the basis of a quality control procedure to define the ultimate use of merchantable cocoa lots as they are received at the factory [32]. In a recent study, we were able to model this sensorial classification by deciphering the volatilome of 206 dark chocolate samples using the DIMS method PTR-MS [32]. This approach, combined with chemometrics and variable selection procedures, allowed us to propose a highly significant prediction model of the sensory categories (poles) based on a limited number of ions (10 to 22 depending on the selection method used) [32]. Some of them were tentatively identified as volatile compounds on the basis of their mass formulae determined thanks to the time-of-flight (TOF) mass analyzer used, and literature data [32]. However, some of these ions represented ion fragments and most of the supposedly molecular ions represented many possible isobaric compounds or isomers. None of them were securely identified and their aroma activities in the chocolates were not determined.

Comparative GC-O studies conducted so far have been aimed at emphasizing key odorants that could contrast samples differing in terms of cultivars, origins, processes, or brands. Distinguishing the same types of samples, only categorized in different classes based on sensory criteria, using a comparative GC-O methodology appears challenging. Therefore, the aim of the present study was to determine the discriminant key aroma compounds that should allow the four previously characterized and modelled sensory poles to be distinguished. To achieve this, three exemplary samples of each sensory pole were chosen on the basis of the availability and respective position in the sensory space of the 206 samples described above. A combined GC-MS and GC-O approach was conducted on each of the selected 12 samples. The key aroma compounds were determined by the detection frequency analysis (DFA) method [33,34], using a panel of 12 assessors. The results obtained for each sample were compared in order to emphasize the discriminant key aroma molecules in the chocolates. Finally, a correspondence analysis (CA) and a hierarchical cluster analysis (HCA) were conducted in order to find potential relationships between the discriminant features and the 12 samples.

## 2. Results

In a previous study [32], four sensory categories (named sensory poles) of dark chocolates, essentially based on aroma evaluation, were characterized and modelled using the headspace fingerprints of 206 samples obtained with a direct introduction mass spectrometry technique (PTR-MS). Key aroma compounds that should discriminate these categories were searched for using 12 samples representative of the sensory poles. The equilibrated positions of these 12 samples (three in each pole) in the sensory space of the 206 chocolates defined in a QDA are illustrated on the principal component analysis (PCA) planes of the sensory data displayed in Figure A1 (Appendix B).

To identify the impact aroma compounds, a GC-O approach, completed by GC-MS, was conducted on the 12 samples by a panel of 12 assessors. The discriminant odorants were determined using a comparative detection frequency analysis (cDFA). The chocolate aroma extracts under study were obtained in triplicate by hydro-distillation under vacuum in a solvent-assisted flavor evaporation (SAFE) device [35], completed by headspace (HS) extracts obtained using solid-phase microextraction (SPME) to account for the most volatile odorants. The reliability and repeatability of the hydro-distillation step were checked by the use of an internal standard. SPME conditions were optimized in order to get the same GC response and odor intensity as in the SAFE method for a volatile “reference” peak (butane-2,3-dione), which was detected by the entire panel in both methods.

### 2.1. Determination of Impact Compounds by GC-O: Comparative Detection Frequency Analysis (cDFA)

The aim of any GC-O experiment is to screen the volatiles isolated from a particular food in order to determine their relative odor potency and to prioritize the most potent odorants, the key odorants, for subsequent identification. Different GC-O methods have been developed within three main paradigms. Dilution techniques (aroma extract dilution analysis, AEDA; and combined hedonic aroma measurement, CHARM) analyze aroma extracts through several successive dilutions as far as no odorant is perceived [36,37]. Both are valuable screening methods that use a large number of dilutions, but generally, a very small number of assessors. Therefore, both methods are not amenable to statistics, and, being based only on individual detection thresholds, they are associated with inherent drawbacks, such as the sensitivity of the small panel of sniffers, with potential inattention and/or specific anosmia, and time-consuming successive sessions [38,39,40,41]. The method named Osme (odor-specific magnitude estimation) is supposed to measure the odor intensity of the eluting species [42]. The third method, detection frequency analysis (DFA), only determines when an odorant is detected [33,34]. Osme requires trained panelists familiar with the odor intensity notion whereas DFA results in an easier task. Both Osme and DFA employ a single optimized extract dilution but a larger number of sniffers; they use the sensitivity of several assessors to average and mitigate inter-individual variation and allow statistical evaluation of the data. Different studies showed that the three methods give similar and correlated results in terms of screening impact odorants [41,43,44,45]. In the present study, DFA was used with a panel of 12 assessors sniffing the 12 samples SAFE extracts (10 assessors for SPME extracts). The triplicate SAFE extracts of each sample were pooled for the GC-O analysis and HS-SPME extracts optimized as described above (see also the experimental section for details).

The GC-O experiments allowed the detection of 8480 odor events all together, which were grouped for each sample in olfactive areas (OAs) on the basis of their linear retention indices’ (LRIs) closeness. Therefore, each OA was characterized by the number of individual odor events detected by the panel members; this number defined its detection frequency, expressed as a percentage as the nasal impact frequency (NIF %) [33]. A detection threshold filter was applied to remove noise and retain only the most intense and significant OAs. No definite rule exists to determine such a threshold. A minimal 12.5% NIF value was set as being necessary to build an OA [46]. However, in the current literature, values varied from this lower level to a 50% NIF threshold [34,47,48]. Owing to the number of odor events detected by the panelists in the 12 samples, illustrating the odorous richness and sensory diversity of the dark chocolates under study, and the number of replicates per sample (12 or 10), a 50% threshold was applied, as also chosen by others [47,48,49]. This meant an OA was finally retained as significant only if its detection frequency was ≥ 50% in at least one sample. By applying this 50% NIF threshold, 96 OAs were finally considered (Table 1), i.e., a rather great number of OAs despite the high threshold level used.

Some specificities were evidenced as some of these OAs were found at high levels in all the samples, whereas some were found more specifically in some samples. However, the total number of significant OAs detected per sample was quite similar, with an average of 43 (min 35–max 53). Thus, among the OAs commonly detected at high levels in all samples and identified as butane-2,3-dione (OA n°7), ethyl 2-methylbutanoate (10), oct-1-en-3-one (19), dimethyltrisulfide (24), trimethylpyrazine (26), 5-ethyl-2,3-dimethylpyrazine and 3-methylthiopropanal (34, coeluted but distinguishable), phenylacetaldehyde (45), 2- and 3-methylbutanoic acid (47), phenylmethanol (59), and 4-vinylguaiacol (80), most of them were previously identified in cocoa mass or dark chocolate or in other cocoa categories (Table 1). Vanillin (OA n°93) was also of this type but was not further considered as vanillin was added in the recipe as a flavoring ingredient. Only a few of these OAs were detected by almost all the assessors (NIF ≈ 100%) in all the samples (Table 1). They are limited to ubiquitous molecules, such as butane-2,3-dione (OA n°7), dimethyltrisulfide (24), 3-methylthiopropanal (34), phenylacetaldehyde (45), 2- and 3-methylbutanoic acid (47), and 4-vinylguaiacol (80). The number of such OAs, not discriminative because they were detected by all the panelists and delivered the highest NIF values in all the samples, was inferior to 10% of the detected OAs, as recommended by Etievant and Chaintreau [41] to allow olfactive discrimination between samples in GC-O, thus validating the extract concentrations chosen for our study [40]. Other OAs were also found in all the samples at common lower levels, for instance, 2-methylpropanal (OA n°2), ethyl propanoate (6), 3-hydroxybutan-2-one (18), 2-ethenyl-6-methylpyrazine (35), acetylpyrazine (43), 1-phenylethyl acetate (48, coeluted with another ester: methyl 2-methylpentanoate), 2,3,5-trimethyl-6-(3-methylbutyl)pyrazine (49), 1*H*-pyrrole-2-carbaldehyde (69), and nonanoic acid (78). Most of them were also previously identified in cocoa or chocolate categories (Table 1). Together with the previous ones, these compounds were found to be important for the overall aroma of the samples and may represent a common aromatic background of dark chocolate, which is not able to discriminate the samples according to their differing sensory properties.

Numerous OAs were found with very different NIF values between samples (Table 1). To consider a significant difference between samples, it was assumed that an NIF value difference of 30% between the lowest and the highest values is at least necessary when working with a panel of eight assessors [33]. By applying this difference threshold for our panel of 12 (or 10 for HS) assessors, an NIF difference > 30% between at least two samples, i.e., a difference of at least 4 assessors (for the SAFE extracts), 73 OAs were considered discriminant among the 96 initially retained (Table 1). Amongst these discriminant odorants, most of them were previously identified as volatile compounds of dark chocolate or cocoa mass (Table 1) and many of them determined as key odorants of dark chocolate. Thus, the Strecker aldehydes 2- and 3-methylbutanal (OAs n°3 and 4) with their characteristic cocoa and chocolate olfactive notes were found together with other key chocolate aldehydes: 2-methylbut-2-enal (OA n°12), heptanal (15), oct-2-enal (30), non-2-enal (39), and 2-phenylbut-2-enal (61) [1,27,29,50,51]. The key alcohols heptan-2-ol (OA n°20), 2-phenylethanol (60), phenol (67), 4-methyphenol (72) [1], butane-2,3-diol (40) [50,52], and guaiacol (58) [29,50] were found to be discriminant together with other alcohols, generally present in cocoa products (Table 1), but for some of them, these have never been described as dark chocolate key aroma compounds: Ethanol (OA n°5), butan-2-ol (8), 2-methylbut-3-en-2-ol (9), 3-methylbutan-1-ol (16), 3-ethoxypropan-1-ol (23), octan-2-ol (28), 3-ethyl-4-methylpentan-1-ol (37), 1-phenylethanol (54), farnesol (88), and octadecan-1-ol (95). Dark chocolate key esters were also found as discriminative features: Ethyl 2- and 3-methylbutanoate (OA n°11 and 12), isoamyl acetate (13), pentyl acetate (14), ethyl phenylacetate (52), 2-phenylethyl acetate (55), and ethyl cinnamate (74). Other esters, for some of them this is the first time being described as key aroma compounds of dark chocolate (Table 1), were also considered as discriminant: Hept-2-yl acetate (OA n°17), ethyl nonanoate (39), 3-hydroxypropyl acetate (or propane-1,3-diol monoacetate, 50), pentan-2-yl benzoate and ethyl dodecanoate (56), methyl tetradecanoate (65), isopropyl palmitate (82), and 2-phenylethyl lactate (86). Some key aroma ketones were also found to be discriminant: Nonan-2-one (OA n°25) already characterized in dark chocolate (Table 1), and others, while cited in cocoa products (Table 1), were described for the first time as key aromas in dark chocolate: Acetophenone (OA n°46) and heptadecan-2-one (81). The hydroxyketone 3-hydroxy-4-phenylbutan-2-one (OA n°84) has never been described in cocoa products but was produced from phenylalanine in the Maillard reaction in a study on roast aroma formation [53]. Discriminant carboxylic acids were also found among dark chocolate key aromas: Acetic acid (OA n°33), butanoic acid (44), and phenylacetic acid (94). Octanoic acid (OA n°71) and dodecanoic acid (92), while found in cocoa products (Table 1), are formally cited for the first time as key odorants of dark chocolate.

Numerous lactones were found to be discriminant, some of them as part of dark chocolate’s key aroma compounds: δ-octenolactone (OA n°66), γ-nonalactone (68), γ-decalactone (77), and δ-decalactone (79); δ-octalactone (63), δ-decenolactone (83), and γ-dodecalactone (89) are new dark chocolate key odorants whereas δ-pentalactone (OA n°53) is newly described in cocoa products. Some important heterocycles were also found amongst the 73 discriminant odorants: The furanones furaneol (4-hydroxy-2,5-dimethylfuran-3(2*H*)-one, OA n°70), a key aroma of dark chocolate; dihydroactinidiolide (4,4,7a-trimethyl-5,6,7,7a-tetrahydro-1-benzofuran-2(4*H*)-one, OA n°87), found as a new dark chocolate key odorant; and 5-[(2*Z*)oct-2-en-1-yl]dihydrofuran-2(3*H*)-one (90), newly described as a key aroma of a cocoa product. Pyrans and pyranones were also part of the discriminant odorants, together with some pyrroles: The pyranol *trans*-linalool-3,7-oxide (6-ethenyl-2,2,6-trimethyltetrahydro-2*H*-pyran-3-ol, OA n°51); the pyranones maltol (OA n°63, 3-hydroxy-2-methyl-4*H*-pyran-4-one), 3-hydroxy-2,3-dihydromaltol (85), newly cited in dark chocolate’s key compounds; dihydromaltol (57), newly determined in a cocoa product; and the pyrroles 2-acetylpyrrole (OA n°64), 5-methyl-1*H*-pyrrole-2-carbaldehyde (73), and 1*H*-indole (91), all three already described as key odorants of dark chocolate (Table 1).

Among the heterocycles, pyrazines were found in numerous key compound examples, and many of them were found to be discriminant. Thus, being already established as key odorants of dark chocolate (Table 1), 2,5-dimethylpyrazine (OA n°21), ethylpyrazine (22), 3-ethyl-2,5-dimethylpyrazine, 2-ethyl-3,5-dimethylpyrazine (32), and 3-isobutyl-2,5-dimethylpyrazine (38) were discriminative features, together with the pyrazines newly identified as key odorants of dark chocolate (Table 1): 2,6-diethylpyrazine (OA n°31), 2-ethenyl-5-methylpyrazine (36), 2-isobutyl-3,5-dimethylpyrazine (38), and 2-isobutyl-3,5,6-trimethylpyrazine (41).

Finally, among the key sulfur aromas, only methanethiol (OA n°1), for the first time being described as a key odorant of dark chocolate, was found to be discriminant to distinguish the 12 samples (Table 1).

Some of these OAs found to be discriminant were characteristic of only one or a few samples. Thus, OAs n°9 (2-methylbut-3-en-2-ol), 63 (δ-octalactone and maltol), 74 (ethyl cinnamate), and 87 (dihydroactinidiolide) reached an NIF value ≥ 50% in sample 1A only. OA n°8 (butan-2-ol) reached this NIF value only in sample 1C, while OAs n°12 (2-methylbut-2-enal), 13 (isoamyl acetate), 32 (3- and/or 2-ethyldimethylpyrazines), and 56 (pentan-2-yl benzoate and ethyl dodecanoate) attained this level only in sample 2C. OAs n°64 (2-acetylpyrrole) and 76 (unknown) seemed characteristic of sample 3C, while OA n°55 (2-phenylethyl acetate) reached this NIF value only in sample 4C. The NIF values of these OAs stayed below 50% in the other samples in rather equilibrated proportions (Table 1). They could be more specific of the respective corresponding samples, where their NIFs reached a value ≥ 50%. Moreover, some OAs could reach an NIF value ≥ 50% in a particular sample, while attaining values just below 50% in other samples. Thus, OA n°77 (γ-decalactone) reached 58% in sample 3A and 42% (i.e., a difference of only two assessors detecting the component) in samples 4B and 4C. When using this type of OA, differentiating samples from sensory poles 3 and 4 will be difficult. On the contrary, some OAs may reflect a strong specificity. Thus, OA n°28 (octan-2-ol) reached a 92% NIF value in sample 4A while being almost not detected in poles 1 and 2 samples. The same applied for OA n°33 (acetic acid), with a value of 100% in sample 1A, while it was not detected in sample 4C. Besides, OA n°52 (ethyl phenylacetate) appeared particular, with a 92% NIF value in sample 1B while it was not detected in the other samples of pole 1 (Table 1).

**Table 1 molecules-25-01809-t001:** Characterization of olfactive areas (OAs) of the 12 chocolate samples (1A to 4C) for which the nasal impact frequency (NIF) ≥ 50% in at least one sample. ^a^ OA number, OA 1 to 10 obtained in HS-SPME-GC-MS, OA from 11 obtained with SAFE extracts; ^b^ Linear retention indices (LRIs) calculated in GC-MS on DB-FFAP and DB-5 columns, “n.d.” = not detected, “-“ = irrelevant; ^c^ NIF values in each sample (%); ^d^ Odor attributes given by the panel; **^e^** Discriminant OA: x = NIF difference > 30% between at least 2 samples, xx = NIF difference > 50% between at least 2 samples (see text); **^f^** Reliability of identification: ***1*** = MS, LRIs and odor identical to published data and/or data found in databases, ***2*** = 1 with additional MW confirmed by CI, ***3*** = 2 with additional injection of standards on equivalent DB-5 and/or DB-FFAP columns, ***4*** = after MDGC-MS/O; **^g^** Compounds identified in references related to cocoa and/or dark chocolate (not exhaustive list) and according to the database Volatile Compounds in Food (VCF): (http://www.vcf-online.nl) [54].

OA ^a^	LRI ^b^ (GC-MS)		NIF ^c^ (%)												Odor Attributes ^d^	Disc.^e^	Compound Identification	Reliability of Identification ^f^	References ^g^
	DB-FFAP	DB-5	1A	1B	1C	2A	2B	2C	3A	3B	3C	4A	4B	4C					
1	701	n.d.	100	100	90	80	90	80	50	70	20	90	60	70	cheese, cabbage, sulfur	xx	methanethiol	*1*	[32,55]
2	817	n.d.	60	60	30	60	60	70	60	60	50	60	50	60	chocolate, cocoa, roasted		2-methylpropanal	*1*	[1,27,28]
3	920	n.d.	60	40	50	60	90	70	50	60	80	80	100	40	cocoa, chocolate	xx	2-methylbutanal	*3*	[1,27,29,50,52]
4	923	n.d.	80	70	60	40	30	50	80	60	40	30	30	60	cocoa	x	3-methylbutanal	*3*	[1,27,28,29,50,51,52]
5	942	n.d.	30	40	40	40	50	30	40	50	30	30	80	40	fruity, solvent	x	ethanol	*3*	[5,11,12,16,56,57,58]
6	961	n.d.	50	50	70	70	50	30	60	40	40	50	60	60	fruity, floral		ethyl propanoate	*1*	[7,54]
7	991 *	n.d.	100	100	100	100	100	100	100	100	90	100	100	100	butter		butane-2,3-dione	*3*	[1,27,29,50,51,52]
8	1025	n.d.	10	10	80	10	10	0	30	20	10	20	20	0	rubber	xx	butan-2-ol	*1*	[7,26,54,59]
9	1040	n.d.	50	20	10	20	10	20	20	0	30	40	20	10	fruity, floral	x	2-methylbut-3-en-2-ol	*2*	[7,56,60]
10	1054 **	n.d.	90	70	60	70	70	60	90	80	80	60	90	60	fruity		ethyl 2-methylbutanoate	*1*	[17,26,29,61]
11	1072	849	33	75	42	50	67	42	67	67	83	42	67	33	fruity, floral	x	ethyl 3-methylbutanoate	*2*	[17,26,29,50,52,62]
12	1108	n.d.	25	25	8	17	42	75	0	8	8	8	8	0	hot plastic	xx	(*E*)-2-methylbut-2-enal	*2*	[1,7,24,27,63]
13	1127	874	8	33	0	17	42	58	8	8	0	17	8	0	fruity, candy	xx	isoamyl acetate	*3*	[29,50,57,60,62,64,65]
14	1183	n.d.	0	33	17	8	50	58	8	8	8	8	8	0	fruity	xx	pentyl acetate	*2*	[12,29,50,52,60,65]
15	1196	903	42	42	17	8	75	50	42	42	42	67	42	17	fruity, floral	xx	heptanal	*3*	[1,27,60,63,65]
16	1211	750	17	33	0	42	50	33	17	25	0	33	50	8	cheesy	x	3-methylbutan-1-ol	*3*	[20,56,57,60]
17 ^$^	1267	891	17	42	0	8	92	75	58	67	75	83	67	42	fruity, flowery	xx	styrene	*2*	[3,12,65]
		1038															hept-2-yl acetate	*2*	[66]
18	1296	742	58	75	83	50	75	50	58	58	75	58	75	50	butter		3-hydroxybutan-2-one	*3*	[1,26,63,67]
19	1309	975	92	83	100	100	100	100	92	75	83	92	100	75	mushroom		oct-1-en-3-one	*2*	[20,22,28,29,51,54,61]
20	1323	902	8	42	17	33	58	50	42	33	25	58	58	42	fruity, mushroom, vegetal	x	heptan-2-ol	*3*	[1,3,26,57,59,60,65,68]
21	1330	914	25	25	17	0	50	33	25	25	25	50	25	8	roasted, chocolate	x	2,5-dimethylpyrazine	*2*	[1,27,28,50,52]
22	1346	916	67	67	50	75	92	75	42	75	67	42	58	42	roasted cereals, peanut	x	ethylpyrazine	*2*	[1,7,24,27,59,67]
23 ^$^	1383	1128	67	0	42	58	17	0	0	0	0	0	0	0	metallic, musty	xx	*allo*-ocimene	*2*	[19,65,68]
		798															3-ethoxypropan-1-ol	*2*	[65]
24	1387	969	83	100	92	92	100	83	83	100	100	100	100	83	sulfur, cabbage		dimethyltrisulfide	*3*	[27,28,29,50,51,52]
25	1397	1091	0	50	0	17	33	42	33	33	67	33	42	25	fruity, floral, vegetal	xx	nonane-2-one	*3*	[6,28,60,64,65]
26	1410	1001	83	92	92	100	92	92	83	100	75	92	100	100	roasted, vegetal, earthy		trimethylpyrazine	*3*	[27,28,29,50,51,52]
27	1419	-	25	58	25	17	58	50	8	8	8	17	8	8	fruity	x	unknown		-
28	1427	n.d.	0	33	0	0	0	25	8	50	42	92	75	67	fruity, floral, candy	xx	octan-2-ol	*1*	[7,11,56,57,65,68]
29	1431	-	67	75	42	92	50	67	58	58	75	75	67	83	roasted, nutty	x	unknown		-
30	1438	1058	67	92	92	92	83	75	92	42	92	100	83	67	vegetal, earthy	xx	(*E*)-oct-2-enal	*1*	[3,16,26,29,68]
31	1440	1075	8	25	17	0	33	33	17	75	17	33	42	33	vegetal	xx	2,6-diethylpyrazine	*2*	[62]
32 ^$^	1450	1076	25	0	25	0	33	50	8	25	8	8	17	17	vegetal, roasted	x	3-ethyl-2,5-dimethylpyrazine and/or 2-ethyl-3,5-dimethylpyrazine	*2*	[27,28]
																			[1,27,29,50,51]
33	1462	n.d.	100	58	92	58	83	67	25	42	67	58	25	0	vinegar	xx	acetic acid	*3*	[1,28,29,50,52]
34 ^$^	1466	1078	100	100	100	100	92	92	100	100	100	100	100	100	roasted, then potato		5-ethyl-2,3-dimethylpyrazine	*2*	[1,50,63,69,70]
		907															3-methylthiopropanal	*3*	[1,27,63,71,72]
35	1497	1016	67	42	42	58	67	58	42	58	67	92	92	50	vegetal, earthy, roasted		2-ethenyl-6-methylpyrazine	*2*	[1,27,62,63,73]
36	1508	1021	0	33	0	0	33	42	33	58	58	67	67	25	vegetal, earthy, roasted	xx	2-ethenyl-5-methylpyrazine	*2*	[54]
37	1514	1004	58	17	58	25	42	17	17	33	58	42	42	17	flowery, vegetal	x	3-ethyl-4-methylpentan-1-ol	*1*	-
38 ^$^	1530	997	50	83	50	75	83	83	75	92	92	83	83	83	vegetal, pepper	x	3-isobutyl-2,5-dimethylpyrazine	*2*	[1,24,27,70]
		1011															and/or 2-isobutyl-3,5-dimethylpyrazine	*2*	[27,70]
39 ^$^	1542	1293	92	92	75	67	75	67	67	75	75	75	58	42	vegetal, cardboard, flowery	x	ethyl nonanoate	*3*	[5,54]
		1159															(*E*)-non-2-enal	*3*	[16,29,51,61,65]
40	1552	792	0	67	8	17	50	50	50	75	67	58	67	50	flowery	xx	butane-2,3-diol	*3, 4*	[5,26,50,52,56,60,65,67]
41	1594	1273	83	33	58	58	50	50	25	25	33	50	25	33	vegetal, cucumber	xx	2-isobutyl-3,5,6-trimethylpyrazine	*2*	[54,70]
42	1597	988	17	25	25	17	25	25	33	50	42	33	50	0	vegetal, earthy		3-hydroxybutanoic acid	*1*	-
43	1635	1063	50	58	67	83	67	67	58	92	58	50	75	58	roasted		acetylpyrazine	*2*	[29,54,65,71,74]
44	1638	788	67	50	50	58	58	92	50	67	58	50	50	33	cheese	xx	butanoic acid	*3*	[16,20,29,51,59,61,68]
45	1653	1046	100	100	100	92	92	83	100	100	100	100	100	92	flowery		phenylacetaldehyde	*3*	[1,27,28,50,51]
46	1660	1066	0	17	8	33	17	17	8	50	33	25	17	33	floral, fruity	x	acetophenone	*3*	[2,60,62,64,65,68,74]
47	1676	886	100	100	92	100	100	100	92	100	100	92	100	92	melted cheese		2-methylbutanoic acid	*3*	[12,29,51,61,67,75]
		876															3-methylbutanoic acid	*3*	[28,29,50,51,52]
48 ^$^	1710	1188	58	50	58	58	58	67	58	67	75	67	67	50	vegetal, roasted, fruity		1-phenylethyl acetate	*2*	[2]
		842															methyl 2-methylpentanoate	*2*	-
49	1726	1386	75	67	67	92	67	67	58	58	83	67	75	83	floral, anise, minty		2,3,5-trimethyl-6-(3-methylbutyl)pyrazine	*2*	[54]
50	1748	955	17	25	8	42	33	33	17	33	33	33	58	50	unpleasant	x	3-hydroxypropyl acetate	*2*	-
51	1766	1175	67	33	0	17	33	50	17	50	50	33	33	33	fruity, roasted, vegetal	xx	*trans*-linalool-3,7-oxide	*2*	[1,12,16,27,50,52,60,63,68]
52	1795	1242	0	92	0	8	75	67	58	50	50	50	42	25	floral	xx	ethyl phenylacetate	*3*	[29,50,51,57,60,62,65,68]
53	1818	958	75	75	58	75	83	83	83	92	67	83	100	75	roasted, vegetal	x	δ-pentalactone	*2*	-
54	1825	1061	67	92	75	50	100	92	92	100	100	92	100	67	floral, rose, fruity	x	1-phenylethanol	*3*	[17,26,56,59,60,65,68]
55	1828	1255	8	17	0	42	25	33	33	33	42	33	42	50	earthy, moldy	x	2-phenylethyl acetate	*3*	[1,28,29,50,51]
56 ^$^	1848	1391	0	33	17	0	17	50	8	33	42	42	17	17	roasted, nut, spicy	x	pentan-2-yl benzoate	*2*	[12]
		1590															ethyl dodecanoate	*3*	[3,56,57,59,76]
57	1869	1064	17	58	75	50	75	83	33	67	83	42	50	58	roasted, caramel, fruity	xx	dihydromaltol	*2*	-
58	1872	1087	58	67	92	67	58	50	83	42	50	67	75	50	roasted, smoked, sweet	x	guaiacol	*3, 4*	[16,20,24,29,50,59]
59	1892	1036	75	67	75	58	83	67	67	67	67	75	75	75	sweet, fruity, floral		phenylmethanol	*2*	[17,20,65,77]
60	1921	1116	67	75	83	100	92	58	83	67	75	67	92	83	floral, rose	x	2-phenylethanol	*3*	[1,27,28,29,50,51]
61	1944	1269	8	25	17	17	58	42	33	42	50	17	42	25	floral	x	2-phenylbut-2-enal	*2*	[1,17,24,26,27,50,57,60,63,65,68]
62	1976	-	50	50	50	58	75	33	67	67	75	58	58	33	roasted, fruity, spicy	x	unknown		-
63 ^$^	1980	1279	67	33	17	0	17	17	0	0	17	0	8	42	fruity, sweet	x	δ-octalactone	*2, 4*	[65,72,77]
		1081															maltol		[1,24,26,28,63,65]
64	1985	1069	17	33	17	17	25	42	33	42	50	42	17	8	hot plastic	x	2-acetylpyrrole	*2*	[24,27,28,57,60,65,67,68]
65	2011	1721	8	50	25	50	25	42	0	25	67	42	25	17	vegetal, metallic	xx	methyl tetradecanoate	*2, 4*	[54,59]
66	2015	1261	25	50	50	42	75	58	50	83	58	42	58	50	sweet, vegetal	xx	δ-octenolactone	*2, 4*	[20,26,29,51,61]
67	2018	982	50	25	0	0	17	25	33	17	50	33	17	33	floral, fruity	x	phenol	*3, 4*	[1,17,24,26,63,68]
68	2039	1358	25	50	33	25	50	25	67	67	58	67	42	33	fruity, sweet	x	γ-nonalactone	*3*	[20,29,54,61,77]
69	2043	1012	67	58	75	58	83	50	67	50	67	50	83	75	sweet, fruity		1*H*-pyrrole-2-carbaldehyde	*2*	[1,7,27,50,59,63,67]
70 ^$^	2046	1063	58	50	17	42	25	67	0	50	0	0	0	58	caramel, strawberry	xx	furaneol	*3, 4*	[1,6,20,24,28,29,51,63]
71	2075	1170	42	50	33	42	42	25	0	42	50	25	33	42	unpleasant	x	octanoic acid	*3*	[5,7,57,59,70,78]
72	2096	1064	42	75	58	83	67	83	83	92	83	83	75	50	animal, unpleasant, urine	x	4-methylphenol	*3*	[1,24,27,29,63,71,77]
73	2122	1124	33	17	25	0	17	25	0	58	42	33	67	42	floral, spicy, fruity	xx	5-methyl-1*H*-pyrrole-2-carbaldehyde	*2*	[28]
74 ^$^	2139	1466	50	17	25	17	33	25	17	17	17	8	8	33	fruity, vegetal	x	(*E*)-ethyl cinnamate	*2*	[1,17,26,29,59,60,63]
75 ^$^	2142	-	25	33	50	33	25	33	17	17	17	8	8	33	floral, sweet, fruity		unknown		-
76 ^$^	2147	-	33	25	8	8	42	8	8	42	58	0	0	0	roasted, spicy	xx	unknown		-
77	2156	1465	33	17	25	17	8	25	58	33	33	17	42	42	sweet, fruity, peach	x	γ-decalactone	*1*	[22,28,51,61,72,79]
78	2197	1265	50	58	33	33	33	42	33	58	50	42	60	58	animal, unpleasant		nonanoic acid	*1*	[5,7,80]
79	2205	1490	17	42	8	0	50	25	50	42	42	17	25	25	fruity, floral, woody	x	δ-decalactone	*2*	[20,22,28,72,77]
80	2212	1308	100	83	100	92	83	100	100	83	100	100	92	92	curry, licorice, clove, spicy		4-vinylguaiacol	*3*	[66]
81	2234	1900	58	58	50	33	83	50	67	92	67	58	92	75	floral, fruity, vegetal	xx	heptadecan-2-one	*2*	[56]
82	2240	2022	0	17	42	25	17	17	33	8	8	8	58	50	floral, fruity	xx	isopropyl palmitate	*1*	-
83	2246	1471	17	0	17	25	33	25	67	42	67	25	0	8	fruity, sweet, coconut	xx	δ-decenolactone	*2*	[20,22,54,77]
84	2272	1343	25	33	17	50	25	33	25	33	50	17	58	58	unpleasant, dust	xx	3-hydroxy-4-phenylbutan-2-one	*2*	-
85	2278	1143	17	25	17	17	17	17	17	50	8	0	33	8	roasted, chicory coffee	x	3-hydroxy-2,3-dihydromaltol	*1*	[54]
86	2338	1501	42	50	25	0	50	50	0	17	8	0	0	0	woody, vegetal	x	2-phenylethyl lactate	*1*	-
87	2353	1527	50	33	17	33	25	25	25	25	8	25	8	8	dust	x	dihydroactinidiolide	*1*	-
88	2365	n.d.	8	8	42	25	33	50	33	17	42	50	42	42	floral	x	farnesol	*1*	-
89	2387	1675	0	8	8	17	0	25	42	25	33	8	58	75	fruity, peach	xx	γ-dodecalactone	*3*	[29,72]
90 ^$^	2412	1652	8	58	17	25	42	58	67	75	50	50	83	75	fruity, floral	xx	5-[(2*Z*)oct-2-en-1-yl]dihydrofuran-2(3*H*)-one	*2*	-
		1214													rubber, medicinal		4-vinylphenol	*3*	[27,29]
91	2464	1290	33	67	25	33	58	67	42	50	42	42	42	58	unpleasant, floral	x	1*H*-indole	*3*	[1,27,29,63]
92	2511	1565	33	42	50	17	42	50	42	50	33	42	25	0	unpleasant, animal, leather	x	dodecanoic acid	*3*	[5,17,26,54]
93 ***	2587	1392	83	92	92	92	92	83	92	83	50	83	83	92	vanilla, sweet, cocoa		vanillin	*3*	[1,27,28,29,51,54]
94	2591	1245	67	42	33	25	42	50	33	50	58	0	0	0	floral, unpleasant	xx	phenylacetic acid	*2*	[26,29,51,54,59,70,77]
95	2602	2086	17	25	58	8	50	17	0	33	42	8	8	17	floral	xx	octadecan-1-ol	*1*	-
96	2628	-	8	8	33	17	0	8	0	8	0	50	25	17	floral		unknown		-

* also identified in the SAFE extract at LRI 995; ** also identified in the SAFE extract at the same LRI; *** vanillin, not further considered (see text); **^$^** coelution not resolved on DB-FFAP column, either deconvoluted or resolved in MDGC-MS-O.

In order to rationalize the data, correspondence analysis (CA) was used to study the potential relationships between the 73 discriminant OAs and 12 samples through the NIF values gathered in Table 1. This multivariate exploratory analysis appeared suitable for the nature of the data that exhibited frequencies of detection. While highly significant (Khi² independence test: *p*-value < 0.0001) and allowing a rather clear separation between groups of samples (Appendix A), the analysis revealed many variables (OAs) that were poorly represented (i.e., localized in the center of the CA plot), exhibiting no real change in the detection frequencies between samples. They represented common key impact compounds but were not able to participate in the differentiation of the different samples (Appendix A). Moreover, a parametric analysis (comparison of k proportions) conducted on the OAs (Khi² test) delivered insignificant *p*-values (α = 0.05) for most of them (Table A1, Appendix C). To remove this noise in the CA, the NIF difference threshold between at least two samples was increased from > 30% to > 50%, meaning that a difference of at least six assessors (for the SAFE extracts) was judged necessary to define a discriminant OA. This more drastic threshold retained 34 discriminant OAs (Table 1) for which most of the *p*-values in the Khi² test of the k proportions comparison were also highly significant (Table 2). Therefore, the selection of the significant variables on the detection frequency basis of the GC-O analyses (difference threshold > 50%) revealed a good accord with the parametric comparison of k proportions. A CA was realized with these 34 significant OAs (Figure 1 and Figure A2, Appendix C), resulting in a highly significant analysis (Khi² independence test: *p*-value < 0.0001), meaning that some relationships between the 34 OAs and the 12 samples should exist. As expected, the center of the CA plots was clarified with fewer ill-represented variables.

CA plots (Figure 1 and Figure A2) were used to study potential proximities between samples on the one hand, and between samples and OAs on the other hand. Factor 1 (36.47% of inertia) clearly separates poles 1 and 2 samples from poles 3 and 4 ones, positioned on the negative and the positive sides of the factor, respectively (Figure 1a). Factor 3 (11.59% of inertia) allows a better separation between poles 3 and 4 samples (Figure 1b). Sample 2A is found in proximity with samples 1A and 1C (on the negative side of F1) while sample 1B is close to samples 2B and 2C, near the center of F1 and on the negative side of F2. These findings were already pinpointed in the related previous experiment, where samples belonging to poles 1 and 2 presented large intra-class distances in a PCA conducted on the samples’ volatilome data [32], a phenomenon also apparent in Figure A1. Meanwhile, samples belonging to poles 3 and 4 were found close together, being obviously very similar in terms of the volatiles composition, as previously noticed [32]. However, as in the PLS-DA previously conducted on the volatilome data [32], samples belonging to poles 3 and 4 were better distinguished on the third factor F3 (Figure 1b). The CA finally clearly distinguished four groups of three chocolates through their proximities on the plots: Sample groups {1A, 1C, 2A}, {2B, 2C, 1B}, {3A, 3B, 3C}, and {4A, 4B, 4C}, respectively. The fact that sample 2A was classified with samples 1A and 1C, and sample 1B was classified with samples 2B and 2C, respectively, illustrated, as already outlined, the large intra-class variability of the corresponding sensory poles 1 and 2, which partially overlapped (Figure A1, Appendix B). The proximity of the sensory poles 3 and 4 with partial overlap was also apparent (Figure A1).

OAs more associated with particular samples are clearly visible on the CA plots (Figure 1 and Figure A2). Thus, OAs n°23 (*allo*-ocimene), 8 (butan-2-ol), 70 (furaneol), 33 (acetic acid), and 41 (2-isobutyl-3,5,6-trimethylpyrazine), found on the negative side of factor F1, distinguish the sample group {1A, 1C, 2A}. Opposed on the positive side of F1, OAs n°28 (octan-2-ol), 36 (2-ethenyl-5-methylpyrazine), 89 (γ-dodecalactone) distinguish group {4A, 4B, 4C}, and to a lesser extent group {3A, 3B, 3C}, together with OAs n°40 (butane-2,3-diol), 90 (5-[oct-2-en-1-yl]dihydrofuran-2(*3H*)-one), 31 (2,6-diethylpyrazine), and to a lesser extent OAs n°17 (hept-2-yl acetate), 25 (nonane-2-one), and 52 (ethyl phenylacetate), seem more specific of group {3}. On the negative side of factor F2, OAs n°12 (2-methylbut-2-enal), 13 (isoamyl acetate), 14 (pentyl acetate), 76 (unknown), and 15 (heptanal) are associated with the sample group {2B, 2C, 1B}, in which OA n°82 (isopropyl palmitate), found on the positive side of F2, is less present. It is noteworthy that, except for OA n°41 related to the sample group {1A, 1C, 2A}, all the OAs with a non-significant *p*-value in the Khi² test of the comparison of k proportions (Table 2) are displayed in the center of the CA plots (Figure 1). They were poorly represented in the correspondence analysis and did not participate in the differentiation of the samples. This was particularly true for the OAs n°44 (butanoic acid, *p*-value 0.463), 66 (δ-octenolactone, *p*-value 0.337), and 84 (3-hydroxy-4-phenylbutan-2-one, *p*-value 0.280). This again revealed a good agreement between both variable selection methods, one based on sensory results inferred from the GC-O difference threshold in detection frequencies, and the other one based on statistics that are more conventional.

In order to go deeper into the data presented in the CA plots and objectively define the relationships that exist between the 34 discriminant OAs and the 12 samples, a heatmap was constructed using the NIF data found in Table 1. This heatmap (Figure 2) independently classified variables (OAs) and individuals (samples) thanks to a hierarchical cluster analysis (HCA) centered on Euclidian distances. The resulting samples’ clustering largely confirmed the correspondence analysis and the evidenced relationships. Thus, four clusters were clearly defined (see also Figure A3, Appendix C): The sample groups were {1A, 1C, 2A} and {1B, 2B, 2C}, reflecting the intra-class variability of sensory poles 1 and 2, as already outlined; and {4B, 4C, 3A, 4A} showing the proximity of sample 3A with pole 4 samples, and particularly with sample 4A, and {3B, 3C}. Four to six clusters of variables could also be clearly seen (Figure 2). The first sample cluster {1A, 1C, 2A} was particularly defined by very low NIF values for a series of compounds grouped together in the HCA. Thus, the low NIFs of OAs n°40 (butan-2,3-diol), 36 (2-ethenyl-5-methylpyrazine), 25 (nonan-2-one), 90 (5-[oct-2-en-1-yl]dihydrofuran-2(3*H*)-one), 28 (octan-2-ol), 17 (hept-2-yl acetate), 15 (heptanal), and 52 (ethyl phenylacetate) characterized this cluster, together with, to a lesser extent, the low NIFs of OAs n°81 (heptadecan-2-one), 73 (5-methyl-1*H*-pyrrole-2-carbaldehyde), 66 (δ-octenolactone), and 31 (2,6-diethylpyrazine). It was also defined by high NIFs of three OAs clustered in an HCA branch: 23 (*allo*-ocimene), 33 (acetic acid), and 41 (2-isobutyl-3,5,6-trimethylpyrazine). The sample group {3B, 3C}, opposed to the first one, was defined by high NIFs of OAs n°94 (phenylacetic acid), 76 (unknown), 51 (linalool-3,7-oxide), and 83 (δ-decenolactone) clustered in an HCA branch, and 95 (octadecan-1-ol) and 57 (dihydromaltol), grouped in another branch. It was also defined by low NIFs of OAs n°1 (methanethiol), 41, 23, and 82 (isopropyl palmitate). The third sample cluster {1B, 2B, 2C} was characterized by medium to high NIFs for the branch grouping OAs n°12 (2-methylbut-2-enal), 13 (isoamyl acetate), and 14 (pentyl acetate); the group 15, 17, and 52; the cluster 1, 70 (furaneol), and 33 (acetic acid); and medium to low NIF values for the cluster 82, 84 (3-hydroxy-4-phenylbutan-2-one), and 89 (γ-dodecalactone). Finally, the last sample cluster {4B, 4C, 3A, 4A} was the less homogeneous, and could be better interpreted by considering the two sub-groups {4B, 4C} and {3A, 4A} defined in the HCA. The first sub-group displayed high NIFs for the OA clusters 73-81, 82-84-89, and for OAs 28 and 90, and medium to high values for OAs 31 and 40. Both sub-clusters shared medium to low values for OA groups 12-13-14, 51-76-94, and for acetic acid (OA 33) and *allo*-ocimene (23). The proximity of samples 3A and 4A in the second sub-group was characterized by medium to low NIFs for cluster dihydromaltol (OA 57)-octadecan-1-ol (OA 95), for furaneol (OA 70) and nonan-2-one (OA 25), and medium to high values for the OA cluster 15-17-52 (Figure 2).

### 2.2. Identification of Impact Compounds

Ninety-six OAs reached the 50% NIF threshold used in the DFA and were considered as significant impact odorants of the chocolate samples under study. Among them, only 28 were defined by a single well-resolved GC-MS peak, and they were easily and unambiguously identified by their EI and CI mass spectra and their LRIs on DB-FFAP by comparison with authentic standard data (Table 1). Some other compounds, although present in co-eluted peaks, displayed clear EI mass spectra, sometimes after deconvolution using the AMDIS or PARADISE [81] software packages. Thus, seven more compounds (3-methylthiopropanal in OA n°34, ethyl nonanoate and non-2-enal in OA 39, 2- and 3-methylbutanoic acid in OA 47, ethyl dodecanoate in OA 56, and 4-vinylphenol in OA 90) could be unambiguously identified (Table 1). For 3-methylthiopropanal and 4-vinylphenol, their respective characteristic odor notes detected by the assessors in the descending part of the GC peaks (potato and medicinal, respectively) also aided their identification. Using the same procedure, 58 compounds were tentatively identified by comparison of their MS, LRI on DB-FFAP, and odor to data found in published literature and/or found in libraries. Injections of the sample extracts on a DB-5 column allowed confirmation of most of the identified peaks after determining their LRIs, which were compared to published data using the column and/or to LRI data found in databases. Among the 93 aroma compounds identified so far (35 unambiguously and 58 tentatively) in 83 OAs, only 17 molecular weights were not confirmed by chemical ionization (CI) using methane and ammonia as reagent gases. CI was a successful method to confirm identification when limited information was present in MS databases and/or when EI mass spectra were ambiguous. For example, MW of OA n°69, tentatively identified by its impure mass spectrum to 1*H*-pyrrole-2-carbaldehyde (MW = 95) on the basis of the similarity index using the Wiley 11th Editition/NIST 2017 database (Figure 3a), was confirmed by methane- and ammonia-CI (Figure 3b).

Thus, the methane-CI spectrum displayed the diagnostic ions [M + H]^+^ at *m*/*z* 96, [M + 29]^+^ = [M + C_2_H_5_]^+^ at *m*/*z* 124, and [M + 41]^+^ = [M + C_3_H_5_]^+^ at *m*/*z* 136. This was confirmed in ammonia-CI by the diagnostic ions [M + H]^+^ at *m*/*z* 96, [M + 18]^+^ = [M + NH_4_]^+^ at *m*/*z* 113, probably enhanced by an impurity found at *m*/*z* 112 in the EI mass spectrum, and [M + 35]^+^ = [M + N_2_H_7_]^+^ at *m*/*z* 130 (Figure 3).

For EI and CI mass spectra data acquisitions, basic/neutral and acidic fractions obtained after chemical fractionation of the chocolate extracts were checked when needed, in order to clarify some co-elutions. For instance, γ-nonalactone (OA n°68), just preceding OA 69 by four LRI units, was more clearly identified in the basic/neutral fraction. Odor descriptions given by the 12 assessors in the DFA experiment were also compared to odor attributes found in databases to aid the identification task. Most of the time, this comparison confirmed the identifications inferred from the MS and LRI data (Table A1, Appendix C). Seven OAs remained problematic in terms of the odor description and/or identification because they exhibited co-eluting species that were clearly visible in EI and CI mass spectra obtained using the DB-FFAP column. Therefore, heart-cutting MDGC-MS/O was used to resolve these problems with the DB-FFAP column in the first dimension and a DB-5 one in the second dimension. Three OAs were thus clearly identified and unambiguously confirmed by MS and LRI data of standards obtained on both column types: Butane-2,3-diol (OA n°40), guaiacol (OA 58), and furaneol (OA 70), with the odor attributes also comparable to published data (Table A1). OA n°63 was tentatively determined as a mixture of δ-octalactone and maltol. As their respective odors, in agreement with the published data, are similar (Table A1), the fruity-sweet note of OA 63 could be due to one of them or to the mixture. Finally, a heart-cut of the OAs n°65, 66, and 67 grouped in a single MDGC run allowed the identification of phenol (67) and a tentative identification of methyl tetradecanoate (65) and δ-octenolactone (66). For OA n°32, it was not possible to differentiate 3-ethyl-2,5-dimethylpyrazine from 2-ethyl-3,5-dimethylpyrazine as these molecules shared the same mass spectra, the same LRIs on both DB-FFAP and DB-5 columns, and the same odor descriptions (Table A1). Moreover, both have been described in dark chocolate (Table 1). Therefore, OA 32 was due to either one of these pyrazines or to a mixture of both volatiles. Finally, within the 96 OAs retained as significant impact components of the dark chocolates under investigation, by applying a 50% SNIF threshold in DFA, 101 odorous compounds were identified (39) or tentatively identified (62) with rather good confidence, and 6 remained as unknown.

## 3. Discussion

The main objective of the study was to identify the most potent odorants in chocolates, and particularly the key odorants that could discriminate the samples, and potentially the predefined sensory poles. Clearly, as usual in GC-O studies, the potent odorants were not the most abundant volatiles in the extracts. Thus, the most abundant compounds found in common in all the samples were acetoin (3-hydroxybutan-2-one, LRI_DB-FFAP_ 1296), trimethylpyrazine (LRI 1410), tetramethylpyrazine (LRI 1480), 3-methylbutanoic acid (LRI 1676), the two diastereoisomers of butane-2,3-diol monoacetate LRIs 1575 and 1587), phenylacetaldehyde (LRI 1653), phenylethanol (LRI 1921), and 2-acetylpyrrole (LRI 1985). In pole 1 samples, acetic acid (LRI 1462) was also found to be particularly abundant. As expected by the powerful aromatic nature of dark chocolate, a large number of odorous compounds were detected by the GC-O panel. Applying a high 50% NIF threshold to the data, 96 olfactive areas were finally retained that revealed 107 active odorants, among which six remained unidentified (Table 1). This rather important retained number, despite the application of a demanding threshold, equals or even surpasses the number of OAs found in highly odorous products, like alcoholic beverages, such as Cognac, for example [40], or even chocolate [29]. Identification of most of the impact compounds were based on classical extract handling and instrumental means, GC-MS in electron and chemical ionization, with the help of chemical fractionation of the extracts and MDGC-MS/O. However, some of them appearing in the co-eluted peaks were tentatively identified by complementary comparison of the odor attributes used by the panel to published odor descriptors. Thus, the odor attributes given by the panel to AO n°17 (fruity, flowery) suggested hept-2-yl acetate, known as fruity, rather than styrene, which imparts a plastic note. The same applied for OA n°23 (metallic, musty) attributed to *allo*-ocimene rather than 3-ethoxypropan-1-ol reported as fruity (Table 2, and Table A1). OA n°90, most often described as fruity and floral, was tentatively attributed to the lactone 5-(oct-2-en-1-yl)dihydrofuran-2(3*H*)-one rather than to 4-vinylphenol only detected by fewer panelists in the descending GC peak as rubber and medicinal (Table 2, and Table A1). OA n°32 was not fully resolved as both candidates 3-ethyl-2,5-dimethylpyrazine and 2-ethyl-3,5-dimethylpyrazine were not separated on DB-FFAP nor on DB5 columns (Table 1) and have both been described with the same vegetal, roasted notes (Table A1). The same applied for OA n°38, attributed to the positional isomer candidates 3-isobutyl-2,5-dimethylpyrazine and 2-isobutyl-3,5-dimethylpyrazine, only separated on the DB5 column but imparting the same vegetal, pepper olfactive note that was not described in consulted databases (Table 1 and Table A1).

Most of the identified key odorants have been found previously in cocoa products, including cocoa mass or liquor, and/or dark chocolate (Table 1). However, to the authors’ knowledge, some of them were described here formally for the first time as key odorants of dark chocolate: Methanethiol (OA n°1), ethanol (5), ethyl propanoate (6), butan-2-ol (8), 2-methylbut-3-en-2-ol (9), 3-methylbutan-1-ol (16), hept-2-yl acetate (17), *allo*-ocimene (23), octan-2-ol (28), 2,6-diethylpyrazine (31), 2-ethenyl-5-methylpyrazine (36), 2-isobutyl-3,5-dimethylpyrazine (38), 2-isobutyl-3,5,6-trimethylpyrazine (41), acetophenone (46), 1-phenylethyl acetate (48), 2,3,5-trimethyl-6-(3-methylbutyl)pyrazine (49), 1-phenylethanol (54), ethyl dodecanoate (56), phenylmethanol (59), δ-octalactone (63), methyl tetradecanoate (65), octanoic acid (71), nonanoic acid (78), heptadecan-2-one (81), δ-decenolactone (83), 3-hydroxy-2,3-dihydromaltol (85), γ-dodecalactone (89), and OA n°92, dodecanoic acid (Table 1, and Table A1). Moreover, to the best of the authors’ knowledge, 13 key odorants are described for the first time in the composition of a cocoa product (Table 1, and Table A1). However, all of them have been previously described in foodstuffs or beverages. Thus, 3-ethyl-4-methylpentan-1-ol (OA n°37) was previously described in brandies [54]; 3-hydroxybutanoic acid (42) was described in various fruits, wine and honey [54]; methyl 2-methylpentanoate (48) in potato and tea [54]; 3-hydroxypropyl acetate (50) in bread and wines [54]; δ-pentalactone (53) in various foods and beverages [54]; dihydromaltol (57) in milk products and wine [54]; isopropyl palmitate (82) in various food products [54]; 3-hydroxy-4-phenylbutan-2-one (84) in honey and wines [54]; 2-phenylethyl lactate (86) in cheddar cheese [54]; dihydroactinidiolide (87) in a lot of foodstuffs, beverages, and seeds [54]; farnesol (88) in a lot of foods and beverages [54]; 5-(oct-2-en-1-yl)dihydrofuran-2(3*H*)-one (90) in chicken [54]; and octadecan-1-ol (95) in a lot of products, including milk products, fruits, and tea [54].

In order to determine the discriminative features that should allow samples to be distinguished, based on the work of Pollien et al. [33], firstly a GC-O comparative approach where a 30% difference threshold was considered in the DFA data, i.e., an NIF difference > 30% between at least two samples, was attempted. Among the initial 96 potent OAs, this procedure revealed 73 OAs in which an NIF difference > 30% between at least two samples exists (Table 1). To understand the discriminative variables better, a correspondence analysis was conducted to visualize the proximities between OAs and samples. Despite its statistical significance, this CA displayed rather noisy plots, where many variables (OAs) poorly represented in the center of the CA plots bore little correspondence information (Appendix A). To look more objectively at the data, a statistical comparison of k proportions (Khi² test) was used on the whole NIF dataset of Table 1. The results clearly confirmed the non-discriminant OAs (*p*-values highly non-significant, α = 0.05) and revealed non-significant *p*-values for most of the OAs ill-defined on the CA plots (Table A1). Therefore, a more demanding difference threshold (50%), i.e., an NIF difference > 50% between at least two samples, was applied to the NIF data. This more drastic difference threshold selected 34 OAs (Table 1) for which most of the *p*-values obtained in the Khi² test were also highly significant (Table 2). These 34 OAs defined by 34 odorants, among which only one remained unknown (Table 2), were considered the discriminative features that allowed the samples to be distinguished. Noteworthy, most of their main odor qualities cited by the panelists corresponded generally to odor attributes that were found in the literature and databases (Table 2). The CA conducted using these 34 key odorants revealed significant proximities between particular odorants and the samples (Figure 1). Finally, the CA distinguished four groups of three samples: {1A, 1C, 2A}, {2B, 2C, 1B}, {3A, 3B, 3C}, and {4A, 4B, 4C}. These groups represented a clear image of the four sensory poles, with each sensory pole being characterized by particular key odorants (see results). These groups and their respective proximities also reflected the intra-class variability of sensory poles 1 and 2 [32] (sample 2A grouped with 1A and 1C, and sample 1B grouped with 2B and 2C, respectively), with the concomitant difficulties encountered in sampling pertinent exemplary chocolates considering their partial overlapping evidenced in Figure A1 and in [32]. They also reflected the similarities of poles 3 and 4 [32], albeit distinguishable (Figure 1). The heatmap produced with the NIF data of the 34 discriminant odorants (Figure 2) largely confirmed the CA. The sample clusters defined by HCA showed the same tendencies: Variability of sensory poles 1 and 2, proximity of poles 3 and 4 with sample 3A grouped with pole 4 samples, and particularly with sample 4A. One advantage of such a heatmap based on HCA is the clustering of explanatory variables, thus evidenced in a better manner. For instance, a cluster of OAs with very low NIF values characterized the samples more related to pole 1 (with also sample 2A), which included high NIF values for a cluster composed of acetic acid (OA n°33), *allo*-ocimene (23), 2-isobutyl-3,5,6-trimethylpyrazine (41), methanethiol (1), and furaneol (70) to a lesser extent. The heatmap appeared complementary to the correspondence analysis for the treatment of GC-O data, with the aim of discriminating chocolate samples differentiated on sensory criteria, with the association of discriminant key odorants. Within these 34 discriminant key odorants, 17 are described formally for the first time as key flavor compounds of dark chocolate (Table 2). The criterion based on the NIF difference threshold introduced by Pollien et al. [33] for discriminating samples in GC-O using a comparative analysis based on the detection frequency (named here cDFA) appeared to be in very good accordance with the statistical approach, which used the Khi² test calculated in the comparison of k proportions (Table 2). However, a few discrepancies were noticed that merit discussion.

Three key odorants out of the 34 retained discriminant ones were not at all significant with *p*-values > 0.15 (Table 2). These compounds, butanoic acid (OA n°44, *p*-value = 0.463), δ-octenolactone (66, *p*-value = 0.337), and 3-hydroxy-4-phenylbutan-2-one (84, *p*-value = 0.280), as already outlined, were situated near the origin in the CA plots, and therefore, were not well represented in the correspondence analysis (Figure 1). They did not belong to the same cluster on the heatmap (Figure 2). However, butanoic acid was found with similar medium NIF values in all the samples except a high value in sample 2C and a low value in chocolate 4C (Table 1); this behavior explained both the retained 50% NIF difference threshold and the non-significant Khi² test. The same applied for δ-octenolactone (high NIF value in the only 3B sample vs. low NIF value in the single 1A one), and to a lesser extent for 3-hydroxy-4-phenylbutan-2-one. Therefore, the three compounds can hardly be considered as discriminant features, as clearly indicated by the Khi² test (*p* > 0.15). A heatmap conducted with the remaining 31 discriminant features using a classical non-specific filtering of 50% on the standard deviation (std) criterion (i.e., eliminating 50% of the variables with the lowest std for clarity purpose) revealed interesting features (Appendix A). Particularly, by removing the non-significant variables (based on the Khi² test) and the variables with the lowest std (both types contributing to background noise), the samples’ clustering appeared in good conformity with the initial sensory classification, with the clusters {3B, 3A, 3C} and {4C, 4A, 4B}, corresponding to sensory poles 3 and 4, well defined on discriminant key odorants (Appendix A).

Besides, seven of the OAs not retained as significant based on the 50% NIF difference threshold had significant *p*-values in the Khi² test (Table A1). Thus, 3-methylbutan-1-ol (OA n°16, *p*-value = 0.030), OA n°27 (unknown, *p*-value = 0.003), 1-phenylethanol (54, *p*-value = 0.002), OA n°56 (pentan-2-yl benzoate/ethyl dodecanoate, *p*-value = 0.035), OA n°63 (δ-octalactone/maltol, *p*-value < 0.001), 2-phenylethyl lactate (86, *p*-value < 0.0001), and OA n°96 (unknown, *p*-value = 0.012) should be considered. It is noteworthy that all these but one (OA n°96) satisfied the 30% NIF difference threshold criterion and were retained in the initial 73 discriminant OAs (Table 1). Their NIF values in the samples were of two types (Table 1): Most of them (six out of seven) had generally low NIF values, with no detection (NIF = 0) in some samples, and were very often characteristic of a particular sensory pole. Thus, OA n°27 was more clearly detected in poles 1 and 2, OA n°63 seemed to characterize pole 1, and 2-phenylethyl lactate (86) was not detected at all in pole 4 and characterized poles 1 and 2, which was contrary to OA n°96 that seemed significantly detected only in pole 4. The remaining 1-phenylethanol (OA n°54) had generally very high NIF values except in one sample (2A). All these behaviors explained both the retained 30% NIF difference threshold and the significant *p*-values in the Khi² test. Therefore, it sounded reasonable to include them as significant variables in the differentiation of the chocolates. A heatmap was calculated using the 38 ‘discriminant’ variables (31 + 7) based on both the NIF difference threshold and Khi² test. For clarity purposes and to highlight the most significant variables that could discriminate the samples, a 50% non-specific filtering on the std criterion was again applied, therefore resulting in only 19 variables being displayed (Figure 4). However, an HCA conducted with the complete set of 38 variables was also performed and resulted in the same sample clustering (Appendix C, Figure A4).

Noteworthy, the grouped samples were respectively gathered in four well-separated clusters {1A, 1C, 2A}, {1B, 2B, 2C}, {3A, 3B, 3C}, and {4C, 4A, 4B} corresponding to the initially defined sensory poles, with the limit of the misclassification of samples 1B and 2A already outlined. Clustered variables allowed qualification of the sample groups (Figure 4). Thus, 5-methyl-1*H*-pyrrole-2-carbaldehyde (OA 73) and octan-2-ol (28) were more perceived in poles 3 and 4 samples. A group of key odorants was clearly less perceived in pole 1 represented by the cluster {1A, 1C, 2A}: γ-dodecalactone (OA 89), OA n°90 5-(oct-2-en-1-yl)dihydrofuran-2(3*H*)-one, butane-2,3-diol (40), 2-ethenyl-5-methylpyrazine (36), hept-2-yl acetate (17), and ethyl phenylacetate (52). Butan-2-ol (OA 8) and δ-decenolactone (83) were more perceived in pole 3. Methanethiol (1) and *allo*-ocimene (23) characterized poles 1 and 2 together with OA n°63 (δ-octalactone/maltol) and furaneol (70) while the last compounds were less perceived in poles 3 and 4. Acetic acid (OA 33) and phenylacetic acid (94) had higher NIF values in poles 1 and 2, phenylacetic acid being particularly absent from pole 4 samples (Table 1). Finally, 2-methylbut-2-enal (OA 12) and 2-phenylethyl lactate (86) were less perceived in poles 3 and 4, the latter characterizing particularly pole 2 chocolates (Figure 4). These 19 particularly significant key odorants that allowed discrimination of the 12 chocolates in agreement with the sensory poles could not be related to the cocoa variety and/or origin as the initial classification was only based on sensory properties [32]. Moreover, the dark chocolate key odorant compounds constitute a flavor balance that is the result of many factors, including the cocoa variety, post-harvest treatments linked to origin, and a complex processing that includes roasting. For instance, acetic acid and phenylacetic acid are final degradation products of the amino acids alanine and phenylalanine, respectively, which accumulate from the fermentation of cocoa beans to the final product [1], but acetic acid is also a marker of the cocoa variety Criollo [31,82]. Heterocyclic compounds like lactones, pyrazines, pyrroles, pyranones, furanones, and the Strecker aldehydes, formed in abundance in the Maillard reaction during the roasting step, are already present in fermented cocoa beans [1,11,17,77,83]. Thus, it was recently reported that interactions between cocoa botanical and geographical origin, formulation, and process showed difficulties in identifying individual markers linked to the different steps along the supply chain [31].

Most of the key odorants identified in the present study were potential candidates for the molecular ions identified in the PTR-ToF-MS analyses of the samples’ headspace volatiles [32]. However, the volatiles with higher molecular weights were only identified in the present study, illustrating the fact that headspace analyses are less sensitive than vacuum extraction procedures. Among the 38 ‘discriminant’ key odorants identified here, only 6 were found in the discriminant ions that allowed classification of the initial 206 chocolate samples [32]: Butan-2-ol (OA n°8), 2-methylbut-2-enal (12), 3-methylbutan-1-ol (16), 2,6-diethylpyrazine (31), butane-2,3-diol (40), and 1-phenylethanol (54). This result reflects the different criteria retained to classify the samples. In the PTR-MS study, the relative abundance of the volatile components (represented by 143 ions) were used after headspace extraction; furthermore, the variables (ions) were highly correlated [32]. In the present study, the discriminative features were determined for their odor quality as key odorants in a comparative GC-O experiment, i.e., in a sequential discrete detection mode. While being impact odorants, they were sometimes found in very low abundance, and a lot of them (with the higher molecular weights) were simply not detected in the samples’ headspace.

The comparative GC-O conducted here used the detection frequency analysis method with the data expressed in nasal impact frequencies. Although this are not a direct measurement of the perceived odor intensities, it can be demonstrated that NIF values increase as a function of the concentration [33], and consequently with odor intensities. It was assumed that working with a panel of 8–10 assessors, an NIF difference of 30% would generally indicate a significant concentration difference for individual perceived odorants [33]. In the present study, we worked with a panel of 10–12 assessors and finally assessed an NIF difference of 50% between at least two samples as being necessary for an odorant to differentiate them with high significance on this component. This assessment was largely confirmed by the Khi² test of comparing k proportions, which was made usable with the type of data obtained using DFA. Therefore, if the NIF values we obtained did not exactly measure the concentrations, at least they gave a good idea of the relative abundances of the key odorants in the samples, which were finally retained as discriminative features.

## 4. Materials and Methods

### 4.1. Chocolate Samples

All the dark chocolates under study were produced in an industrial pilot plant using the same ‘standard’ transformation process, with the same mass of cocoa (65%) from different origins and varieties, of sugar, of soy lecithin, and of vanillin. They were supplied by the Valrhona Company, chocolate producer in Tain L’Hermitage (France). Twelve chocolates, three in each of the four sensory poles previously defined at the industrial level, were chosen among 206 chocolate samples that were used to build a predictive model [32]. Being representative of the sensory categories, they were chosen according to four decision criteria: Availability (sufficient quantity available to conduct all the experiments of the project), uniform distribution in the four sensory poles, coverage of the maximum acceptable variability within each sensory pole, and distinct origins. Their positions in the sensory space are highlighted in the PCA planes of the sensory data of the 206 samples in Figure A1 (Appendix B) for illustrative purposes. In the following, they will be noted xA, xB, and xC, x being the sensory pole (x = 1 to 4). The samples were stored under vacuum at −20 °C before their analysis.

### 4.2. Extraction of the Volatiles

After being thawed at room temperature, each sample of dark chocolate was cut into small cubes (ca. 1 cm^3^). Suspended in 100 mL of ultra-pure water (MilliQ system, Millipore, Bedford, MA, USA), the sample was placed in the sample flask of a solvent-assisted flavor evaporation (SAFE) glassware [35], where, together with a magnetic stirrer, 300 µL of an aqueous solution at 93 mg/L of 2-methylheptan-3-one (CAS 13019-20-0; 99% pure; Sigma-Aldrich, St Louis, MO, USA) used as internal standard were added. The resulting standard concentration was 0.28 mg/L. The round-bottom flask was placed in a water bath at 37 °C (just above the chocolate melting point) and the stirred slurry with melted chocolate was distilled under vacuum in the SAFE apparatus at 1 Pa. After a distillation time of 1h30min, the frozen hydro-distillate (ca. 100 mL) was thawed at room temperature, and then a liquid-liquid extraction was conducted with methylene chloride (CH_2_Cl_2_) as solvent (Carlo Erba, Val de Reuil, France; purity > 99.9%, distilled just before use). Three successive 15-min extraction steps were realized under agitation using 3 × 15 mL CH_2_Cl_2_ in a water-ice bath (ca. 4 °C) and the recovered organic extracts were pooled and dried over anhydrous Na_2_SO_4_ (5 g). The extract was then filtered through glass wool before being concentrated to 400 µL (adjusted volume with CH_2_Cl_2_) using two successive Kuderna-Danish apparatuses (Merck, Darmstadt, Germany) of decreasing size equipped with a Snyder column. The extracts obtained in triplicate for each chocolate sample were stored at −20 °C before use.

### 4.3. Determination of Impact Compounds by GC-O Comparative Detection Frequency Analysis (cDFA)

The 12 chocolate extracts (pooled triplicates of each extraction) were submitted to GC-O using the detection frequency analysis (DFA) method [33,34]. The extracts’ concentration was optimized in dummy assays conducted with three assessors to follow the recommendation of Etiévant and Chaintreau [41] to avoid overexpressing the number of odorants that could be detected by all the panelists (thereafter not discriminant).

Analyses were performed on a 6890A GC (Agilent Technologies, Santa Clara, CA, USA) equipped with an FID and an in-house sniffing port using a DB-FFAP column (30 m × 0.32 mm i.d., 0.5 mm film thickness; Agilent Technologies). He4e, 1 µL of extract was injected using a splitless/split injector in splitless mode for 0.5 min, then switched to split mode (25 mL/min) at a temperature of 240 °C. The initial oven temperature was set to 40 °C and then increased at 4 °C/min to a final temperature of 240 °C held for 10 min. Analyses were performed in constant flow mode at a carrier gas (He) velocity of 44 cm/s. At its end, the column was connected to a Y-type seal glass, and the effluent was split into two equal parts (50% to FID, 50% to sniffing port) by two deactivated capillaries (both 1.1 m, 0.32 mm i.d.). The FID and transfer line to the sniffing port were heated at 240 °C. Humidified air (25 mL/min) was added to the transfer line to prevent nasal mucosa dehydration. Linear retention indices (LRIs) were calculated by a weekly injection of a reference solution of n-alkanes (C_7_ to C_30_; Sigma-Aldrich) according to van den Dool and Kratz [84].

Twelve assessors belonging to the CSGA staff (8 women and 3 men, 21 to 61 years old, nine of them with previous experience in GC-O) participated after having been informed and having signed a consent form. Each of them sniffed the 12 extracts once, in a randomized order using a Williams Latin square design, for a period of ca. 40 min starting 3.8 min after the injection (solvent delay). Data were acquired by the OpenLab software (6850/6890 GC System, V2.3, Agilent Technologies) for the chromatographic part and by the ODP recorder (Gerstel, Mülheim an der Ruhr, Germany) for the descriptors citing part (a button and a microphone allowed the recording of odor events and their vocal description). DFA is based on the determination of olfactive areas (OAs) in a sample by gathering all the odor events detected by the panel on the basis of their LRI closeness, grouped if the difference is inferior to a few LRI values. A threshold of 50% for the detection frequency, also known as the nasal impact frequency (NIF) [33], was set as necessary to retain an OA [34,48]. This threshold equals a minimum of six odor events detected by the panel in a sample to retain an OA, i.e., an OA was retained only when six assessors detected it in at least one sample [34,48].

cDFA was performed to obtain a first impression of the odorants, which may contribute to the overall aroma of the dark chocolates and to highlight differences between them. Although they are not a direct measurement of the perceived odor intensities, NIFs increase with concentration [33]. Therefore, they can be used to compare peak intensities between aromagrams. According to Pollien et al. [33], a difference in NIF values of at least 30% (between the lowest and the highest values of one OA) is assumed to be a significant concentration difference. Therefore, to be considered as a discriminant OA, a 30% difference (corresponding to a difference of four odor events) between at least two samples for the given OA was applied. To highlight very discriminant OAs, a 50% difference (six differing odor events between at least two samples) was also applied in a second time.

To determine the very volatile impact compounds whose retention times do not allow separation of them from the solvent peak on the DB-FFAP column (generally for LRIs ≤ 1000), a headspace (HS) technique was used. A solid-phase microextraction (SPME) method was optimized (chocolate sample size, addition of water or not, equilibration time and temperature, extraction time, desorption time, type of SPME fiber) in order to get the same GC response and odor intensity as in the SAFE method for a volatile reference peak, clearly identified as butane-2,3-dione (LRI: 995 on DB-FFAP, odor descriptor: butter) and detected by the entire panel (NIF: 100%). Thus, 2 g of chocolate cut in small cubes (ca. 0.5 cm^3^) were suspended in 1 mL of purified water within a 20-mL sample vial containing a magnetic stirrer. The vial, closed by a PTFE-lined screw cap, was equilibrated under agitation (250 rpm) at 60 °C for 15 min in a water bath. Then, the extraction was realized with a triple-phase divinylbenzene/carboxen/polydimethylsiloxane (DVB/CAR/PDMS) SPME 2-cm fiber (Supelco Sigma-Aldrich) for 30 min at the same temperature. Then, the SPME fiber was desorbed for 5 min in the GC injector maintained at 240 °C (splitless mode). As only the most volatile compounds were sniffed in that case, the GC oven set at an initial temperature of 40 °C was programmed at 4 °C/min to 80 °C and then to 240 °C (maintained for 10 min) at 20 °C/min. Other GC and signal acquisition parameters were the same as the ones mentioned earlier, except the sniffing period that lasted 5 min only, and 10 assessors from the initial 12 ones participated (two were not available). As previously stated, a weekly injection of n-alkanes was used for LRI calculation, this time after adsorption on the same SPME fiber. The GC-O data were processed the same way as previously stated.

All through the GC-O procedure, the quality of the GC column was checked for repeatability (retention times, peak heights and peak areas) weekly by injecting a reference solution (Grob Test Mix, Sigma-Aldrich).

### 4.4. Identification of the Impact Compounds

The compounds responsible for OAs were identified by GC-MS.

The triplicate SAFE extracts of the 12 chocolate samples were analyzed on a 7890A GC coupled to a 5975C mass selective detector (MSD, Agilent Technologies) using the same column as in the GC-O study. GC-MS data of SPME extracts were also obtained in duplicate on the DB-FFAP column, using the same conditions as those used for the GC-O experiments. A complementary study was performed on a DB-5MS column (30 m × 0.32 mm i.d., 0.5 µm film thickness, Agilent Technologies) to confirm the identifications by obtaining MS and LRIs on a second column with a different polarity, and thus avoiding overlooking possibly coeluting compounds. The used GC conditions were the same. The data were obtained on the DB-5 column on a pooled solution of the triplicate SAFE extracts of the 12 chocolates. Analyses were conducted using the same chromatographic parameters with a solvent delay of 3.5 min, except for SPME analyses, and LRIs were calculated as previously described. Electron ionization (EI) spectra were obtained with electron energy of 70 eV at a rate of 4 scans/s, covering the *m*/*z* range 29–350 with a source temperature of 230 °C. Data were acquired using the ChemStation software (ver. A.03.00, Agilent Technologies). The reliability of the compound identification was first assured by comparison of the experimental mass spectra to mass spectral data contained in various databases: NIST 2017/Wiley 11th Edition, MassBank (https://masbank.eu/MassBank), Pherobase (https://www.pherobase.com), and our in-house database INRAMass containing more than 10,000 mass spectra of volatiles. The software packages AMDIS (ver. 2.73, NIST) and PARADISE [81] (ver. 2.92, http://www.models.life.ku.dk/paradise) were used for mass spectra deconvolution of coeluted peaks. Besides spectral information, compound identification was confirmed by comparison of the experimental LRIs to published data and to data found in the following online databases: NIST Chemistry WebBook (http://webbook.nist.gov/chemistry), Volatile Compounds in Food (http://www.vcf-online.nl) [54], the Pherobase, and the LRI & Odour database (http://www.odour.org.uk). When standards were available in our collection of aroma compounds, identifications were confirmed by comparing their MS and LRI obtained on equivalent DB-FFAP and/or DB-5 columns.

Chemical ionization (CI) was also carried out with methane and ammonia as reagent gases on the pooled triplicates of each sample. CI analyses were conducted with a source pressure of 0.1 kPa for both gases at a source temperature of 150 °C and with an electron energy of 240 eV. Molecular weights (MWs) were determined by observing diagnostic ions depending on chemical classes [85,86].

To aid compound identification, a chemical fractionation of the pooled triplicate extracts of each chocolate sample was also performed to separate the basic/neutral fraction from the acidic one. An aliquot (200 µL) of each CH_2_Cl_2_ extract was diluted in 100 mL of purified water. The aqueous solution was adjusted to pH 9 with NaOH (0.045 M) and agitated for one hour. The basic/neutral fraction was recovered by extraction with CH_2_Cl_2_ (3 × 10 mL). The remaining aqueous solution was adjusted to pH 2 with aqueous HCl (18%), stirred for one hour, and the acidic fraction was recovered by extraction with CH_2_Cl_2_ (3 × 10 mL). Both organic fractions were dried, filtered, and concentrated as previously described, and analyzed by GC-MS on both DB-FFAP and DB-5 columns.

Finally, to separate some co-eluting species not clearly resolved by the use of the columns of different polarities, a two-dimensional GC-MS/O system (MDGC-MS/O) was used. The first gas chromatograph (GC1) was a 7890A GC (Agilent Technologies) equipped with FID as a monitoring detector and a DB-FFAP column (30 m × 0.25 mm i.d., 0.5 µm film thickness, Agilent Technologies). The second GC (GC2) was also a 7890A GC equipped with a DB-5MS column (30 m × 0.25 mm i.d., 0.5 µm film thickness, Agilent Technologies) and coupled to a 5975C MSD (Agilent Technologies) and to a sniffing port (ODP 3, Gerstel). The connection between GC1 and GC2 was provided by a Deans switch (Agilent Technologies) followed by a cryotrap system (CTS, Gerstel) cooled down to −100 °C by liquid nitrogen. Fractions transported by the Deans switch (heart-cuts) from GC1 to GC2 were released to GC2 by a rapid heating (ca. 20 °C/s) of the CTS trap to 240 °C. GC ovens were successively temperature programmed from 40 to 240 °C at a rate of 4 °C/min. All other parameters were fixed as previously described. After the second column separation in GC2, 2/3 of the flow was diverted to the ODP and 1/3 to the MSD via two deactivated capillaries of adequate dimensions (0.83 m × 0.18 mm i.d. and 0.50 m × 0.10 mm i.d., respectively) via a capillary flow purged splitter (Agilent Technologies).

### 4.5. Statistical Data Analysis

All the statistical data treatments were performed using the software packages XLSTAT (Addinsoft, Paris, France) and/or Statistica (ver. 13.3, TIBCO Software Inc., Tulsa, OK, USA).

## 5. Conclusions

The aim of the present study was to identify the discriminant key odorants that should allow four previously characterized sensory categories of dark chocolates to be distinguished, which were modelled using the volatilome of 206 samples [32]. To address the question, a GC-O study was conducted by 12 assessors using a comparative detection frequency analysis (cDFA) on 12 samples chosen on availability and exemplariness criteria. A nasal impact frequency (NIF) difference of 50% for a key odorant between at least two samples was retained to differentiate the samples. A correspondence analysis (CA) revealed a classification that could be related to the sensory categories initially defined, through the proximities found between the most discriminant key odorants and the chocolate samples. The approach was confirmed and completed by a statistical analysis (Khi² test on proportions) made feasible with the DFA data. Finally, 38 key odorants discriminated the samples and allowed retrieval of the sensory categories thanks to a hierarchical cluster analysis (HCA). The discriminative relationships were illustrated in a heatmap, where the 19 most significant key odorants were identified.

## Figures and Tables

**Figure 1 molecules-25-01809-f001:**
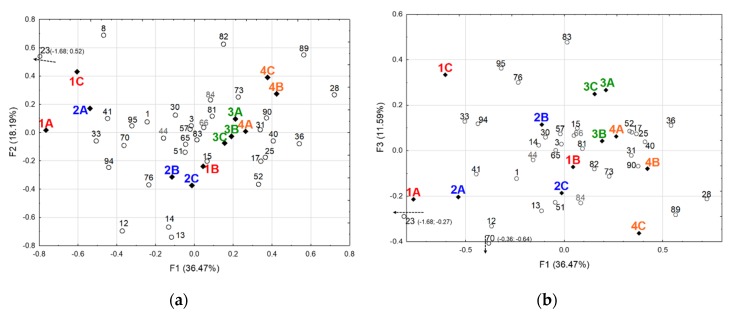
Correspondence analysis (CA) between the 12 samples and the 34 discriminant OAs defined by their NIF values. OAs (light circles) are plotted according to their NIF in samples (plain diamonds) in the dimensions 1 and 2 that gathered 54.66% of cumulative inertia (**a**), and 1 and 3 (**b**), respectively. The CA plots are zoomed in for clarity, and the coordinates of extra variables (23 and 70) indicated in brackets with their direction. The OA numbers are those found in Table 1. The sample names are colored for illustrative purpose, with pole 1 samples appearing in red, pole 2 samples in blue, pole 3 samples in green, and pole 4 ones in orange. A 3D plot (dimensions 1, 2, and 3) of the CA may be found in Appendix C (Figure A2). CA independence test: Khi² = 5444 (critical value 408, α = 0.05, degrees of freedom = 363), *p* < 0.0001.

**Figure 2 molecules-25-01809-f002:**
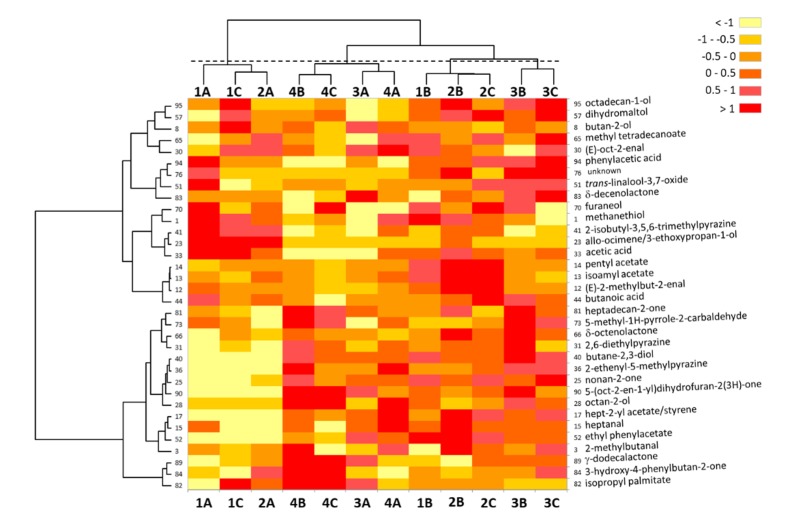
Heatmap displaying the results of a hierarchical cluster analysis (HCA) conducted independently on both samples’ and variables’ (OAs) dimensions, for the 34 discriminant OAs. NIF values’ importance varies from >1 (highest value, in red) to < −1 (lowest values, in yellow). OA numbers are those found in Table 1. An HCA conducted on only samples showing the 4 distinctive clusters displayed here may be found in Appendix C (Figure A3) for clarity purposes. The data were centered and scaled; dissimilarity Euclidian distances were used with the Ward amalgamation method.

**Figure 3 molecules-25-01809-f003:**
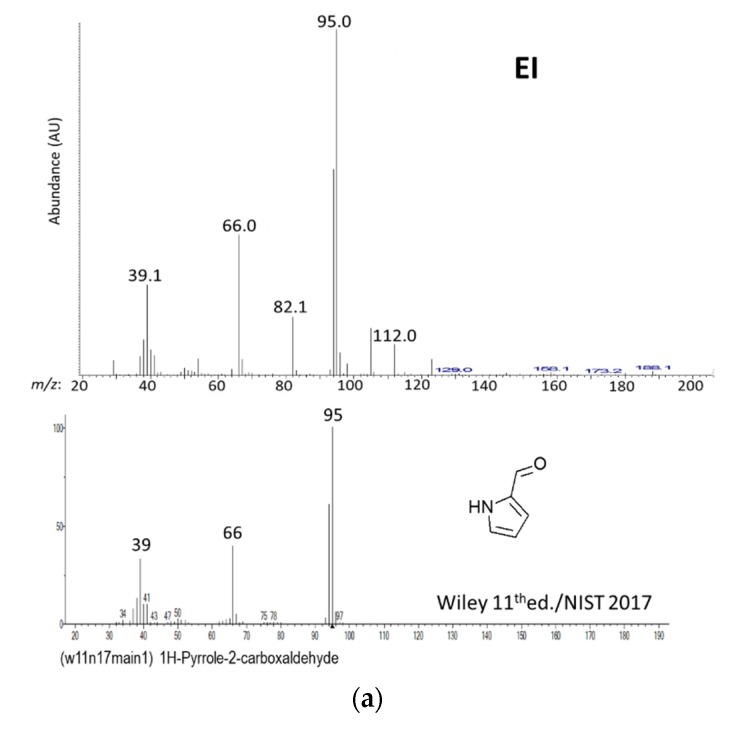
Mass spectra of 1H-pyrrole-2-carbaldehyde (OA n°69, MW 95) obtained in EI compared to the Wiley 11th Edition/NIST 2017 database reference spectrum (**a**) and in CI with methane and ammonia as reagent gases (**b**). Diagnostic ions on both CI spectra are indicated (**b**).

**Figure 4 molecules-25-01809-f004:**
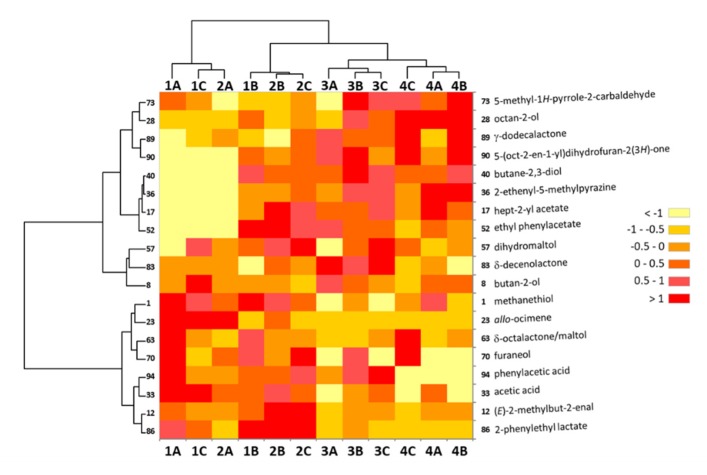
Heatmap displaying the results of a hierarchical cluster analysis (HCA) conducted independently on both samples and variables (OAs) dimensions, for the 38 “discriminant” OAs determined by both NIF difference threshold and khi² test (see text). NIF values importance varies from >1 (highest value, in red) to < −1 (lowest values, in yellow). OA numbers are those found in Table 1. The data were centered and scaled; dissimilarity Euclidian distances were used with the Ward amalgamation method; %(std) non-specific filtering was used with a 50% threshold, resulting in the display of the only 19 variables with the highest std. An HCA conducted on the samples with the 38 variables, showing the same four distinctive clusters displayed here, may be found in Appendix C (Figure A4) for clarity purposes.

**Table 2 molecules-25-01809-t002:** Key discriminant aroma compounds that characterize the four sensory poles as determined in a cDFA GC-O analysis using a 50% discriminative threshold (see text for a complete explanation).^a^ OA number, as in Table 1; ^b^ LRI on DB-FFAP, as in Table 1; ^c^ Odor attributes given by the panel; ^d^ Identification (refer to Table 1); ^e^ Chemical Abstracts Service registry number; ^f^ Mass formula; ^g^ Molecular Weight; ^h^ Pertinent odor attributes found in the databases VCF and The Good Scents Company (http://www.thegoodscentscompany.com/); ^i^
*p*-value of the Khi² test (α= 0.05) obtained for the parametric test comparison of k proportions using the data of Table 1.

OA ^a^	LRI ^b^	Odor ^c^	Identification ^d^	CAS ^e^	Formula ^f^	MW ^g^	Lit. Odor ^h^	*p*-value ^i^
1	701	cabbage, sulfur	**methanethiol ***	74-93-1	CH_4_S	48.1	cabbage, sulfur	<0.001
3	920	cocoa, chocolate	2-methylbutanal	96-17-3	C_5_H_10_O	*86.1 ***	cocoa, nutty	0.081
8	1025	rubber	**butan-2-ol ***	78-92-2	C_4_H_10_O	74.1	medicine, solvent	<0.001
12	1108	hot plastic	(*E*)-2-methylbut-2-enal	1115-11-3	C_5_H_8_O	*84.1*	solvent, ethereal	<0.0001
13	1127	fruity, candy	isoamyl acetate	123-92-2	C_7_H_14_O_2_	*130.2*	fruity, banana	<0.001
14	1183	fruity	pentyl acetate	628-63-7	C_7_H_14_O_2_	*130.2*	fruity, banana	<0.001
15	1196	fruity, floral	heptanal	111-71-7	C_7_H_14_O	*114.2*	fresh, green	0.037
17	1267	fruity, flowery	**hept-2-yl acetate ***	5921-82-4	C_9_H_18_O_2_	*158.2*	fruity	<0.0001
23	1383	metallic, musty	***allo*-ocimene ***	673-84-7	C_10_H_16_	*136.2*	herbal, peppery	<0.0001
25	1397	fruity, floral, vegetal	nonan-2-one	821-55-6	C_9_H_18_O	*142.2*	sweet, herbal, fruity	0.020
28	1427	fruity, floral, candy	**octan-2-ol ***	123-96-6	C_8_H_18_O	130.2	fruit, fresh, green	<0.0001
30	1438	vegetal, earthy	(*E*)-oct-2-enal	2548-87-0	C_8_H_14_O	126.2	green, herbal, leaf	0.017
31	1440	vegetal	**2,6-diethylpyrazine ***	13067-27-1	C_8_H_12_N_2_	*136.2*	green	0.011
33	1462	vinegar	acetic acid	64-19-7	C_2_H_4_O_2_	*60.1*	vinegar, pungent	<0.0001
36	1508	vegetal, earthy, roasted	**2-ethenyl-5-methylpyrazine ***	13925-08-1	C_7_H_8_N_2_	*120.1*	coffee	<0.001
40	1552	flowery	butane-2,3-diol	513-85-9	C_4_H_10_O_2_	*90.1*	floral	<0.001
41	1594	vegetal, cucumber	2-isobutyl-3,5,6-trimethylpyrazine *	46187-37-5	C_11_H_18_N_2_	*178.3*	-	0.107
44	1638	cheese	butanoic acid	107-92-6	C_4_H_8_O_2_	*88.1*	cheese	0.463
51	1766	fruity, vegetal, roasted	*trans*-linalool-3,7-oxide	39028-58-5	C_10_H_18_O_2_	*170.2*	floral, woody, wintergreen	0.061
52	1795	floral	ethyl phenylacetate	101-97-3	C_10_H_12_O_2_	*164.2*	floral	<0.0001
57	1869	caramel, fruity, roasted	**dihydromaltol ***	38877-21-3	C_6_H_8_O_3_	*128.1*	-	0.019
65	2011	vegetal, metallic	**methyl tetradecanoate ***	124-10-7	C_15_H_30_O_2_	*242.4*	orris, petal, waxy	0.020
66	2015	sweet, vegetal	δ-octenolactone	16400-69-4	C_8_H_12_O_2_	*140.2*	coconut ^$^	0.337
70	2046	caramel, strawberry	furaneol	3658-77-3	C_6_H_8_O_3_	*128.1*	caramel, strawberry	<0.0001
73	2122	floral, spicy, fruity	5-methyl-1*H*-pyrrole-2-carbaldehyde	1192-79-6	C_6_H_7_NO	*109.1*	-	0.005
76	2147	roasted, spicy	unknown	-	-	-	-	<0.001
81	2234	floral, fruity, vegetal	**heptadecan-2-one ***	2922-51-2	C_17_H_34_O	*254.5*	-	0.076
82	2240	floral, fruity	**isopropyl palmitate ***	142-91-6	C_19_H_38_O_2_	298.5	fatty, oily	0.010
83	2246	fruity, sweet, coconut	**δ-decenolactone ***	54814-64-1	C_10_H_16_O_2_	*168.2*	coconut, fruity	<0.001
84	2272	unpleasant, dust	**3-hydroxy-4-phenylbutan-2-one ***	5355-63-5	C_10_H_12_O_2_	*164.2*	burnt plastic	0.280
89	2387	fruity, peach	**γ-dodecalactone ***	2305-05-7	C_12_H_22_O_2_	*198.3*	peach, fruit	<0.0001
90	2412	fruity, floral	**5-[(2*Z*)oct-2-en-1-yl]dihydrofuran-2(3*H*)-one ***	156318-46-6	C_12_H_20_O_2_	*196.3*	-	0.001
94	2591	floral, unpleasant	phenylacetic acid	103-82-2	C_8_H_8_O_2_	*136.1*	floral, urine	<0.001
95	2602	floral	**octadecan-1-ol ***	112-92-5	C_18_H_38_O	270.5	oily	0.010

***** although most of them previously identified in cocoa products (see Table 1), to the authors’ knowledge, these compounds (in bold character) are formally described for the first time as key aroma compounds of dark chocolate; ******** MW in bold italic means MW confirmed by CI; ^$^ odor description in [22]; -: not described or not relevant.

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
