# Peer review of "Key Aroma Compounds of Dark Chocolates Differing in Organoleptic Properties: A GC-O Comparative Study"

_molecules, 2020, doi:10.3390/molecules25081809_

Round 1

Reviewer 1 Report

The manuscript investigated the key aroma compounds of dark chocolates by GC-O combined with multiple analysis methods. It is a well-planned study, concerning a very interesting topic. Strength of the article, data processing and interpretation are sufficient and comprehensive. However, some major modifications need to be done before the manuscript is completely accepted. My detailed comments are as follows:

  1. Title

It would be better if “GC-O” replaced the “GC-Olfactometry” in the title.

  1. Abstract

Care should be more taken for the writing of significant results and major conclusions.

  1. Introduction

The purpose or reason for the research being reported, and its significance, originality, or contribution to new knowledge in the field should be clearly and concisely stated.

  1. Results

It is recommended that the sections be refined to make the thinking clearer. Please delete some unnecessary explanations.

  1. Line 648-660

The “Chocolate samples” section should be rewritten because the illogicality is not so good.

  1. Line 712-722

It is necessary to rewrite and simplify the content.

  1. Conclusions

Some of the sentences in the conclusion are repeated with the abstract. Please replace these repeated statements and write brief.

  1. There are many grammatical errors in the present manuscript. Please check the full text and modify them. What’s more, so much needless repetition makes the manuscript long-winded.

Author Response

The authors would like to thank the reviewer for the positive comments and for the remarks to improve the manuscript.

Here are our response (italicized) to the reviewer's comments:

  1. Title

It would be better if “GC-O” replaced the “GC-Olfactometry” in the title.

done

  1. Abstract

Care should be more taken for the writing of significant results and major conclusions.

We do agree that the first version of the abstract did not report significant results and major conclusions in the most pertinent way. We amended this in the revised version.

  1. Introduction

The purpose or reason for the research being reported, and its significance, originality, or contribution to new knowledge in the field should be clearly and concisely stated.

We have modified the introduction accordingly, especially in the last part where the objectives to conduct this original research are better stated.

  1. Results

It is recommended that the sections be refined to make the thinking clearer. Please delete some unnecessary explanations.

Accordinly, this part has been simplified by removing some unnecessary explanations, and checking for needless repetitions.

  1. Line 648-660

The “Chocolate samples” section should be rewritten because the illogicality is not so good.

The paragraph has been rearranged accordingly, in order to present the sampling in a more logical way.

  1. Line 712-722

It is necessary to rewrite and simplify the content.

The paragraph has been rewriten and the content simplified, as suggested.

  1. Conclusions

Some of the sentences in the conclusion are repeated with the abstract. Please replace these repeated statements and write brief.

We do not agree with the remark. Moreover, as it is usual, conclusions are supposed to emphasize the main findings, which have also to be stressed in the abstract (see your remark on the abstract). The purpose of an abstract and of conclusions is not the same, and it is not surprising to find the same information. Moreover, our conclusions are brief.

8. There are many grammatical errors in the present manuscript. Please check the full text and modify them. What’s more, so much needless repetition makes the manuscript long-winded.

Although the reviewer did not underline the grammatical errors in the manuscript, a complete grammatical check, as well as a complete spell check required by the second reviewer, have been done. Moreover, a special care has been done to remove any needless repetition in the revised manuscript.

Reviewer 2 Report

In this work, the authors perform an experimental investigation aimed to the identification of the odor-active compounds for the discrimination of dark chocolate samples as a continuation of their previous findings. Experiments were conducted by means of the combination of gas chromatography – mass spectroscopy (GC-MS) and gas chromatography – olfactometry (GC-O) with comparative detection frequency analysis (cDFA) implemented by nasal impact frequency (NIF).

The main findings concern the discrimination of key odorants, some of which for the first time, establishing the characteristic aroma of dark chocolate. In fact, although 8480 odor events were detected by GC-O experiments, they were grouped in main olfactive areas (OAs),   by evaluating the so-called Linear Retention Indices (LRI), with a detection threshold, imposed by the analyses performed with Nasal Impact Frequency (NIF %), of 50%.

The strengths of the paper lie also in the execution of the statistical analysis that in terms of correspondence analysis (CA), studying the potential relationships between the 73 discriminant OAs and the 12 samples through the NIF values, and of the hierarchical cluster distribution, allowed a deeper comprehension of the results also by means of a proper heatmap that finally helped to validate the results about the identification of nineteen most significant key odorants.

Being not an expert on odorants, I would not enter into the details and significance of the discriminating compounds, however I have to stress that the work is well written, the results are sound and of interest for the people working in the field.

Therefore, in my opinion the manuscript can be accepted for publication in Molecules.

Author Response

The authors would like to thank the reviewer for the very positive remarks.